# AI-based histopathology image analysis reveals a distinct subset of endometrial cancers

Amirali Darbandsari[1,9], Hossein Farahani [2,3,9], Maryam Asadi[2,9], Matthew Wiens[2], Dawn Cochrane[4], Ali Khajegili Mirabadi [2], Amy Jamieson[5], David Farnell[3,6], Pouya Ahmadvand[2], Maxwell Douglas[4], Samuel Leung[4], Purang Abolmaesumi[1], Steven J. M. Jones [7], Aline Talhouk[5], Stefan Kommoss[8], C. Blake Gilks[3,6], David G. Huntsman[3,4,10], Naveena Singh[3,6,10,11], Jessica N. McAlpine [5,10] & Ali Bashashati [2,3,10] ✉

Endometrial cancer (EC) has four molecular subtypes with strong prognostic value and therapeutic implications. The most common subtype (NSMP; No Specific Molecular Profile) is assigned after exclusion of the defining features of the other three molecular subtypes and includes patients with heterogeneous clinical outcomes. In this study, we employ artificial intelligence (AI)-powered histopathology image analysis to differentiate between p53abn and NSMP EC subtypes and consequently identify a sub-group of NSMP EC patients that has markedly inferior progression-free and disease-specific survival (termed 'p53abn-like NSMP'), in a discovery cohort of 368 patients and two independent validation cohorts of 290 and 614 from other centers. Shallow whole genome sequencing reveals a higher burden of copy number abnormalities in the 'p53abn-like NSMP' group compared to NSMP, suggesting that this group is biologically distinct compared to other NSMP ECs. Our work demonstrates the power of AI to detect prognostically different and otherwise unrecognizable subsets of EC where conventional and standard molecular or pathologic criteria fall short, refining image-based tumor classification. This study's findings are applicable exclusively to females.

The clinicopathological parameters used for decades to classify endometrial cancers (EC) and guide management have been sub-optimally reproducible, particularly in high-grade tumors[1,2]. Specifically, inconsistency in grade and histotype assignment has yielded an inaccurate assessment of the risk of disease recurrence and death. As a result, many women affected by EC may be over-treated or are not directed to treatment that might have reduced their risk of recurrence. In 2013, the Cancer Genome Atlas (TCGA) project demonstrated that endometrial cancers could be stratified into four distinct prognostic groups using a combination of whole genome and exome sequencing,

[1]Department of Electrical and Computer Engineering, University of British Columbia, Vancouver, BC, Canada. [2]School of Biomedical Engineering, University of British Columbia, Vancouver, BC, Canada. [3]Department of Pathology and Laboratory Medicine, University of British Columbia, Vancouver, BC, Canada. [4]Department of Molecular Oncology, British Columbia Cancer Research Institute, Vancouver, BC, Canada. [5]Department of Obstetrics and Gynaecology, University of British Columbia, Vancouver, BC, Canada. [6]Vancouver General Hospital, Vancouver, BC, Canada. [7]Michael Smith Genome Sciences Center, British Columbia Cancer Research Center, Vancouver, BC, Canada. [8]Department of Women's Health, Tübingen University Hospital, Tübingen, Germany. [9]These authors contributed equally: Amirali Darbandsari, Hossein Farahani, Maryam Asadi. [10]These authors jointly supervised this work: David G. Huntsman, Naveena Singh, Jessica N. McAlpine, Ali Bashashati. [11]Deceased: Naveena Singh. ✉e-mail: ali.bashashati@ubc.ca

microsatellite instability (MSI) assays, and copy number analysis[3]. These subtypes were labeled according to dominant genomic abnormalities and included 'ultra-mutated' ECs harboring *POLE* mutations, 'hypermutated' identified to have microsatellite instability, copy-number low, and copy-number high endometrial cancers.

Inspired by this initial discovery, our team and a group from the Netherlands independently and concurrently developed a pragmatic, clinically applicable molecular classification system that classifies ECs into: (i) *POLE* mutant (*POLE*mut) with pathogenic mutations in the exonuclease domain of *POLE* (DNA polymerase epsilon, involved in DNA proofreading repair), (ii) mismatch repair deficient (MMRd) diagnosed based on the absence of key mismatch repair proteins on immunohistochemistry (IHC), (iii) p53 abnormal (p53abn) as assessed by IHC, and (iv) NSMP (No Specific Molecular Profile), lacking any of the defining features of the other three subtypes[4,5]. Categorization of ECs into these subtypes recapitulates the survival curves/prognostic value of the four TCGA molecular subgroups and enhances histopathological evaluation, offering an objective and reproducible classification system with strong prognostic value and therapeutic implications. In 2020, the World Health Organization (WHO) recommended integrating these key molecular features into standard pathological reporting of ECs when available[6].

*POLE*mut endometrial cancers have highly favorable outcomes with almost no deaths due to disease. While the three other molecular subtypes are associated with more variable outcomes (MMRd and NSMP are considered 'intermediate risk' and p53abn ECs have the worst prognosis), within each subtype there are clinical and prognostic outliers[7–10]. This is particularly true within the largest subtype, NSMP (representing ~50% of ECs). The majority of NSMP tumors are early-stage, low-grade, estrogen-driven tumors likely cured by surgery alone. However, a subset of patients with NSMP EC experience a very aggressive disease course, comparable to what is observed in patients with p53abn ECs. At present, limited tools exist to identify these aggressive outliers and current clinical guidelines do not stratify or direct treatment within NSMP EC beyond using pathologic features[11,12]. Thus, for half of diagnosed endometrial cancers, i.e., NSMP EC, the assumption of indolence is inappropriate and clinicians need tools for accurate risk stratification of individual patients when making treatment decisions.

With the rise of artificial intelligence (AI) in the past decade, deep learning methods (e.g., deep convolutional neural networks and their extensions) have shown impressive results in processing text and image data[13]. The paradigm-shifting ability of these models to learn predictive features from raw data presents exciting opportunities with medical images, including digitized histopathology slides[14–17]. In recent years, these models have been deployed to reproduce or improve pathology diagnosis in various disease conditions (e.g.,[18–20]), explore the potential link between histopathologic features and molecular markers in different cancers including EC[17,21–24], and directly link histopathology to clinical outcomes[25–28]. More specifically, three recent studies have reported promising results in the application of deep learning-based models to identify the four molecular subtypes of EC from histopathology images[22,23,29].

In this work, building on a recent study reporting morphological heterogeneity in NSMP ECs and nuclear features typical of p53abn ECs in some tumors of this subtype[30], we built a deep learning-based image classifier to differentiate between the NSMP and p53abn ECs. We then hypothesized that within the NSMP molecular subtype of endometrial cancer, there is a subset of patients with aggressive disease whose tumors have histological features similar to p53abn EC and that these tumors can be identified by deep learning models applied to hematoxylin & eosin (H&E)-stained slides. Our results show that these cases (referred to as *p53abn-like NSMP*) have inferior outcomes compared to the other NSMP ECs, similar to that of p53abn EC, in three independent cohorts. Furthermore, shallow whole genome sequencing studies suggested that the genomic architecture of the *p53abn-like NSMP* differs from other NSMP ECs, showing increased copy number abnormalities, a characteristic of p53abn EC.

## Results

### Patient cohort selection and description

2318 H&E-stained hysterectomy tissue sections from 1272 patients with histologically confirmed endometrial carcinoma of NSMP or p53abn subtypes were included in this study[3–5]. Our discovery cohort included 155 whole-section slides (WSI) from 146 patients from TCGA[3] and 431 WSIs (222 patients) from Tubingen University[5]. Two separate validation Canadian cohorts were included in this study: (1) the British Columbia (BC) cohort, a tissue microarray (TMA) dataset corresponding to 290 patients from our own center[4], and (2) Cross Canada (CC) cohort, 640 WSIs (614 patients) collected from 26 hospitals across Canada[31]. Tables 1–3 show the clinicopathological features of the discovery and validation cohorts. Overview of the cohorts, outcome information, treatment information, and distribution of samples in the CC cohort across different centers can be found in Supplementary Tables 1–4.

**Table 1 | Clinicopathologic features of the discovery set**

| Variable | Total | NSMP | p53abn |
|---|---|---|---|
| Total | 363 | 268 (73.83%) | 95 (26.17%) |
| **Age at diagnosis** | | | |
| < 60 yrs | 121 (33.33%) | 110 (41.04%) | 11 (11.58%) |
| ≥ 60 yrs | 242 (66.67%) | 158 (58.96%) | 84 (88.42%) |
| **Histotype** | | | |
| Endometrioid | 288 (79.34%) | 262 (97.76%) | 26 (27.37%) |
| Non-endometrioid | 75 (20.66%) | 6 (2.24%) | 69 (72.63%) |
| **Tumor grade** | | | |
| Low grade (G1–2) | 258 (71.07%) | 246 (91.79%) | 12 (12.63%) |
| High grade (G3) | 105 (28.93%) | 22 (8.21%) | 83 (87.37%) |
| **FIGO stage** | | | |
| I-II | 291 (80.17%) | 239 (89.18%) | 52 (54.74%) |
| III-IV | 71 (19.56%) | 28 (10.45%) | 43 (45.26%) |
| Unknown | 1 (0.28%) | 1 (0.37%) | 0 |

**Table 2 | Clinicopathologic features of the BC validation set**

| Variable | Total | NSMP | p53abn |
|---|---|---|---|
| Total | 288 | 193 (67.01%) | 95 (32.99%) |
| **Age at diagnosis** | | | |
| < 60 yrs | 81 (28.13%) | 72 (37.70%) | 9 (9.47%) |
| ≥ 60 yrs | 205 (71.18%) | 119 (62.30%) | 86 (90.53%) |
| Unknown | 2 (0.69%) | 2 (1.04%) | 0 |
| **Histotype** | | | |
| Endometrioid | 195 (67.71%) | 172 (89.12%) | 23 (24.21%) |
| Non-endometrioid | 91 (31.60%) | 19 (9.84%) | 72 (75.79%) |
| Unknown | 2 (0.69%) | 2 (1.04%) | 0 |
| **Tumor grade** | | | |
| Low grade (G1–2) | 151 (52.43%) | 146 (75.65%) | 5 (5.26%) |
| High grade (G3) | 137 (47.57%) | 47 (24.35%) | 90 (94.74%) |
| **FIGO stage** | | | |
| I-II | 216 (75.00%) | 166 (86.01%) | 50 (52.63%) |
| III-IV | 69 (23.96%) | 24 (12.44%) | 45 (47.37%) |
| Unknown | 3 (1.04%) | 3 (1.55%) | 0 |

## Histopathology-based machine learning classifier to differentiate NSMP and p53abn ECs

Fig. 1 depicts our AI-based histopathology image analysis pipeline. A subset of 27 whole-section H&E slides from the TCGA cohort were annotated by a board-certified pathologist (DF) using a custom in-house histopathology slide viewer (*cPathPortal*) to identify areas containing tumor and non-tumor cells (myometrium, endometrial stroma, and benign endometrial epithelium). A deep convolutional neural network (CNN)-based classifier was then trained to acquire pseudo-tumor and benign annotations for the remaining slides in the discovery cohort. The identified tumor regions were then divided into $512 \times 512$ pixel patches at 20x objective magnification. The number of extracted patches from each subtype and performance measure for the tumor stroma classifier can be found in Supplementary Tables 1 and 5, respectively. To address variability in slide staining due to differences in staining protocols across different centers, and inter-patient variability, we utilized the Vahadane color normalization technique[32]. We then trained a VarMIL model[33] based on multiple instance learning (MIL) to differentiate H&E image patches associated with p53abn and NSMP ECs.

In a "group 10-fold" cross-validation strategy, the patients in our discovery cohort were divided into 10 groups, and in various combinations, 60% were used for training, 20% for validation, and 20% for testing; resulting in 10 different binary p53abn vs. NSMP classifiers. These 10 classifiers were then used to label the cases as p53abn or

### Table 3 | Clinicopathologic features of the CC validation set

| Variable | Total | NSMP | p53abn |
|---|---|---|---|
| **Total** | 614 | 416 (67.75%) | 198 (32.25%) |
| **Age at diagnosis** | | | |
| < 60 yrs | 199 (32.41%) | 168 (40.38%) | 31 (15.66%) |
| ≥ 60 yrs | 415 (67.59%) | 248 (59.62%) | 167 (84.34%) |
| Unknown | | | |
| **Histotype** | | | |
| Endometrioid | 419 (68.24%) | 380 (91.35%) | 39 (19.7%) |
| Non-endometrioid | 195 (31.76%) | 36 (8.65%) | 159 (80.3%) |
| Unknown | | | |
| **Tumor grade** | | | |
| Low grade (G1–2) | 390 (63.52%) | 376 (90.38%) | 14 (7.07%) |
| High grade (G3) | 199 (32.41%) | 38 (9.13%) | 161 (81.31%) |
| Unknown | 25 (4.07%) | 2 (0.48%) | 23 (11.62%) |
| **FIGO stage** | | | |
| I-II | 487 (79.32%) | 369 (88.7%) | 118 (59.6%) |
| III-IV | 127 (20.68%) | 47 (11.3%) | 80 (40.4%) |
| Unknown | 0 | 0 | 0 |

NSMP and their consensus was used to come up with a label for a given case. For patients with multiple slides, to prevent data leakage between training, validation, and test sets, we assigned slides from each patient to only one of these sets.

Fig. 2A and Supplementary Table 6 show the receiver operating characteristics (ROC) and precision/recall curves as well as performance metrics of the resulting classifiers for the discovery and BC validation cohorts, respectively. These results suggest that our p53abn vs. NSMP classifier achieves 89.4% and 79.8% mean balanced accuracy (across the 10 classifiers) and area under the curve (AUC) of 0.95 and 0.88 in both the discovery and BC validation sets, respectively (for details see Supplementary Tables 7 and 8 and Supplementary Fig. 1).

### Identification of a subset of NSMP ECs with inferior survival

Our proposed ML-based models classified 17.65% and 20% of NSMPs as p53abn for the discovery and validation cohorts, respectively (Supplementary Table 6). These cases (referred to as *p53abn-like NSMP* group) presumably show p53abn histological features in the assessment of H&E images even though immunohistochemistry did not show mutant-pattern p53 expression and these were therefore classified as NSMP by the molecular classifier. We hypothesized that such cases may in fact exhibit similar clinical behavior as p53abn ECs.

Fig. 2B, C show the progression free survival (PFS) and disease-specific survival (DSS) of the discovery and BC validation sets. Compared to the rest of the NSMP cases, *p53abn-like NSMP*s had markedly inferior PFS (10-year PFS 55.7% vs. 89.6% ($p = 2.7e-7$)) and DSS (10-year DSS 62.6% vs. 93.7% ($p = 1.8e-7$)) in our discovery cohort. These findings were confirmed in the BC cohort, with 20% of the 195 patients categorized as p53abn-like tumors, showing 10-year PFS of 65.4% vs. 91.2% ($p = 1.1e-4$) and DSS of 58.3% vs. 84.3% ($p = 5.3e-5$). In addition, a comparison of the PFS and DSS between *p53abn-like NSMP* and p53abn ECs revealed a trend, though not statistically significant, in which *p53abn-like NSMP*s had better outcomes compared to p53abn ECs in both the discovery and BC validation cohorts (Supplementary Fig. 2A, B).

Of note, our model also identified a subset of p53abn ECs (representing 20%; referred to as *NSMP-like p53abn*) with a resemblance to NSMP as assessed by H&E staining. While we observed marginally superior disease-specific survival in the identified cases compared to the rest of the p53abn group both in the discovery and BC validation cohorts, progression free survival was not significantly different between the groups (Supplementary Fig. 3A, B).

### Robustness of *p53abn-like NSMP* subtype

Our proposed deep learning-based model was built to differentiate between NSMP and p53abn EC subtypes. Given that these subtypes are determined based on molecular assays, their accurate identification from routine H&E-stained slides would have removed the need to perform molecular testing that might only be available in specialized centers. However, our observation of imperfect results and

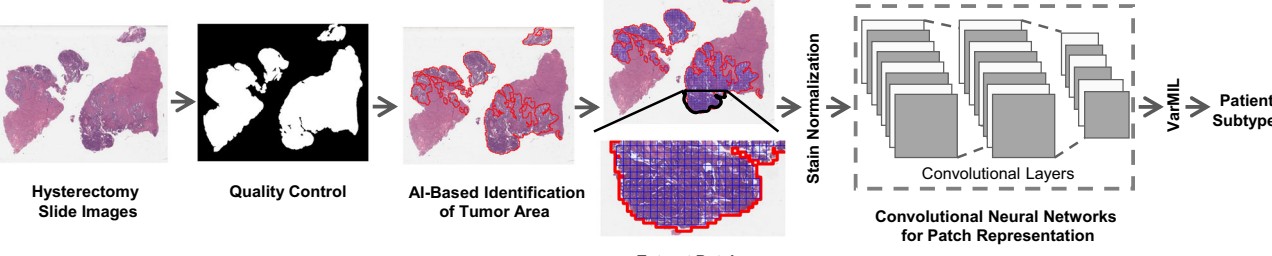

**Fig. 1 | Workflow of the AI-based histopathology image analysis.** First, the quality control framework, HistoQC[81], generates a mask that comprises tissue regions exclusively and removes artifacts. Then, an AI model to identify tumor regions within histopathology slides is trained. Next, images are tessellated into small patches and normalized to remove color variations. The normalized patches are fed to a deep-learning model to derive patch-level representations. Finally, a model based on multiple instance learning (VarMIL) was utilized to predict the patient subtype.

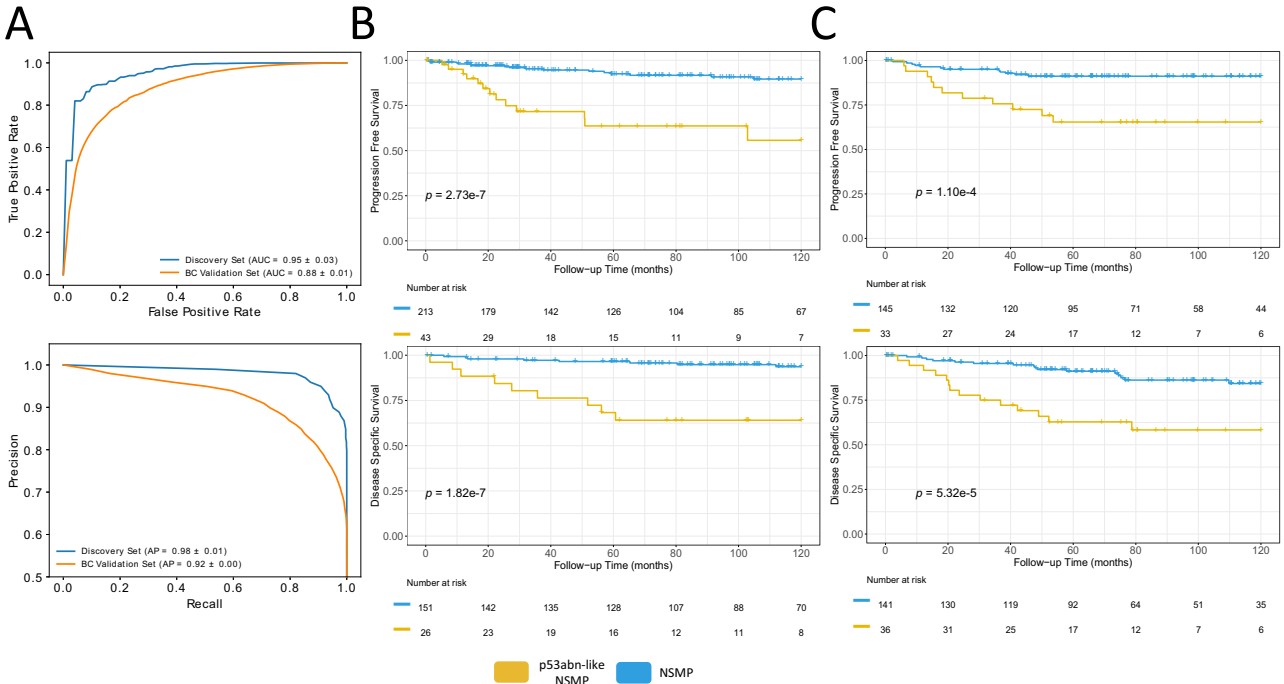

**Fig. 2 | Performance statistics and Kaplan Meier (KM) survival curves for AI-identified EC subtypes. A** AUROC and precision-recall plots of average of 10 splits for p53abn vs. NSMP classifier for discovery and validation sets, (**B**) KM curves associated with PFS and DSS (where available) for the discovery set, using a two-sided log-rank statistical test, (**C**) KM curves associated with PFS and DSS (where available) in the BC validation set, using a two-sided log-rank statistical test (Source data are provided as a Source Data file).

characterization of discordant cases as *p53abn-like NSMP* required further investigation to rule out the possibility of a superior deep learning model that could result in a better performance in differentiating p53abn and NSPMP molecular subtypes. Therefore, we implemented seven other deep learning-based image analysis strategies including more recent state-of-the-art models to test the stability of the identified classes (see Methods section for further details). Our results showed that these models also achieve balanced accuracies ranging from 83.5–95% and 77.3–80.2% and AUCs ranging from 0.88–0.98 and 0.8–0.88 in both the discovery and validation sets, respectively (Supplementary Fig. 4 and Supplementary Tables 9, 10). Furthermore, Kaplan-Meier survival analysis of the so-called *p53abn-like NSMP* group identified by these models also corroborated with our initial findings in which this subgroup had statistically significant inferior survival compared to the rest of the patients (Supplementary Fig. 5). These results suggest that the choice of the algorithm did not substantially affect the findings and outcome of our study. To further investigate the robustness of our results, we utilized an unsupervised approach in which we extracted histopathological features from the slides in our validation cohort utilizing KimiaNet[34] feature representation. Our results suggested that *p53abn-like NSMP* and the rest of the NSMP cases constitute two separate clusters with no overlap (Fig. 3A) suggesting that our findings could also be achieved with unsupervised approaches. It is noteworthy to mention that we utilized the original KimiaNet weights for feature extraction without any finetuning the model on our datasets. To assess the sensitivity of the unsupervised approach to the choice of dimensionality reduction technique, we experimented with DenseNet121[35], Swin[36], and ResNet50. The analysis revealed that identified clusters remain consistent (i.e., two clusters) across these techniques (Supplementary Fig. 6).

## Comparison of NSMP and *p53abn-like NSMP*

We compared various clinical, pathological, and molecular variables to investigate further the differences between NSMP and *p53abn-like*

*NSMP* cases (Supplementary Tables 11–13). Our analysis showed an enrichment of *p53abn-like NSMP* cases with higher grade and higher stage tumors ($p = 1.4e\text{-}25$; $p = 2.4e\text{-}4$, respectively). In a multivariate Cox regression analysis, the association between *p53abn-like NSMP* and progression free survival remained significant in the presence of grade, stage, and histology ($p = 0.01$ and Hazard Ratio = 2.5; Supplementary Table 14). Furthermore, Fig. 3B shows enrichment for estrogen receptor (ER) and progesterone receptor (PR) positive cases in the *p53abn-like NSMP*s (compared to NSMP cases that were classified as NSMP by AI) in the subset of the cohort that the status of these markers were available ($p = 5.2e\text{-}3$ and $p = 2.3e\text{-}4$, respectively).

## Independent pathology review of selected NSMP cases

Two expert gynepathologists (NS, CBG) independently reviewed whole section slides of a subset of NSMP cases including the *p53abn-like NSMP* subtype. They specifically assessed whether tumors showed nuclear features that have been previously described as being associated with *TP53* mutation/mutant pattern p53 expression in endometrial carcinoma[30]. The *p53abn-like NSMP* cases were enriched with tumors showing increased nuclear atypia, as assessed by altered chromatin pattern, nucleolar features, pleomorphism, atypical mitoses, or giant tumor cells ($p = 0.00005$ for both reviewers).

## Genomic characterization of *p53abn-like NSMP* cases

We next sought to investigate the molecular profiles of *p53abn-like NSMP* cases in our BC validation set for which we had access to tissue material. Targeted sequencing of exonic regions in a number of genes (more details in the Targeted point mutation profiling section of the Methods) revealed enrichment of *p53abn-like NSMP* cases with *TP53* mutations (Fisher's exact test p-values = 3.14e-4; Fig. 3B). More specifically, we identified eight (out of 39) *p53abn-like NSMP* tumors that had normal p53 IHC results (hence classified as NSMP by ProMisE classifier) but in fact harbored *TP53* mutations by sequencing. These cases are examples of the well-known phenomenon of normal p53

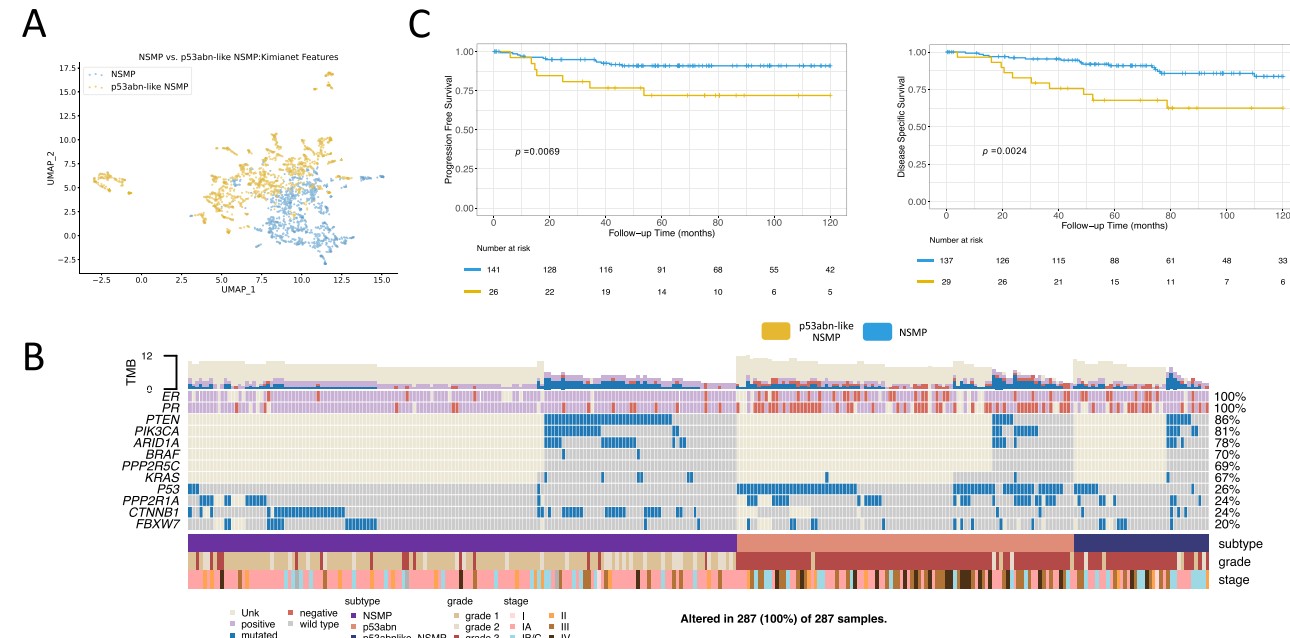

**Fig. 3 | KimiaNet features, clinicopathological features, point mutations, and KM curves of the validation cohort. A** Histopathological features from the slides in the validation cohort utilizing KimiaNet feature representation from the slides in the validation cohort demonstrate that *p53abn-like NSMP* and the rest of the NSMP cases constitute two separate clusters, (**B**) Clinicopathological features and point mutation data for the BC validation cohort, (**C**) KM curves associated with PFS and DSS for the BC validation cohort after removing TP53 mutant cases, using a two-sided log-rank statistical test (Source data are provided as a Source Data file).

protein levels despite there being a pathogenic mutation, which occurs in <5% of cases. However, even after removing these eight *TP53* mutant cases, the worse prognosis of *p53abn-like NSMP* tumors persisted (Fig. 3C). Our ML model, therefore, identifies tumors with false negative immunostaining for p53, i.e., they lack mutant pattern protein expression despite having a *TP53* mutation, but also identifies a subset of NSMP cases with features of p53abn morphology by H&E but neither mutation pattern immunostaining nor a mutation in sequencing TP53, and these have inferior survival compared to the rest of the NSMP cases.

We next selected representative samples of NSMP, p53abn, and *p53abn-like NSMP* cases in our validation cohort and performed shallow whole genome sequencing (sWGS). Overall, copy number profile analysis of these cases revealed that *p53abn-like NSMP* cases harbor a higher fraction of altered genome compared to NSMP cases but still lower than what we observe in p53abn cases (Fig. 4A; *p* = 0.035). These findings were further validated in the TCGA cohort (Fig. 4B; *p* = 5.46e-5).

We next investigated the gene expression profiles associated with the *p53abn-like NSMP*, NSMP, and p53abn tumors within the TCGA cohort. While majority of the p53abn and NSMP groups were clustered separately, unsupervised analysis of the gene expression profiles did not reveal any differences between *p53abn-like NSMP* group and other subtypes, i.e., they did not have a unique gene expression profile but instead clustered with one of the known molecular subtypes (Fig. 4C). We then performed pairwise differential expression analysis and pathway analysis, separately comparing *p53abn-like NSMP* and p53abn groups against NSMP cases. These results suggested the upregulation of PI3k-Akt, Wnt, and Cadherin signaling pathways both in *p53abn-like NSMP* and p53abn groups (compared to NSMP). Interestingly, while these pathways were up-regulated in both groups, we found little to no overlap between the specific down- and up-regulated genes in the *p53abn-like NSMP* and p53abn groups (compared to NSMP) suggesting that the molecular mechanisms associated with p53abn and p53abn-like tumors might be different even though p53abn and *p53abn-like NSMP* groups had similar histopathological profiles as assessed based on H&E slides.

## Validation of *p53abn-like NSMP* subtype in a multi-center dataset

Variability in tissue processing and data collection across different centers and hospitals is known to introduce inconsistencies in the appearance of histopathology slides. While expert human pathologists can adapt to visual color variability between stained slides, AI-based diagnostic models trained on digitized pathology slides from one center may face challenges in generalizing to data collected from other centers[37]. We next evaluated the generalizability of our proposed AI-based p53abn vs. NSMP classifier on a dataset collected from 26 hospitals across Canada (CC cohort). Our proposed models achieved a balanced accuracy and AUC of 66.3–88.5% and 0.88–0.95, respectively (Supplementary Tables 6, 7 and Supplementary Fig. 4) and classified 6.25% of NSMP cases as *p53abn-like NSMPs*. Similar to the discovery and BC validation cohorts, the *p53abn-like NSMP* subtype in the CC cohort had inferior PFS and DSS (Fig. 4D, 5-year PFS 62.52% vs. 88.92% (*p* = 5.41e-6)) and DSS (5-year DSS 66.60% vs. 99.39% (*p* = 8.20e-13)), suggesting that the proposed AI-based classifier is generalizable to datasets collected from other centers.

## Impact of *p53abn-like NSMP* on risk group assignment

We next sought to determine whether the finding of *p53abn-like* status by AI, in an NSMP endometrial carcinoma, would potentially change the risk group category i.e., if the tumor was classified as if it were p53abn molecular subtype rather than NSMP, would that impact on the final risk group assignment. The ESGO/ESTRO/ECP 2021[38] risk group classification is based on molecular subtype, stage, histotype, grade, lymphovascular invasion, and the presence of residual disease; this risk group (Low, Intermediate, High-intermediate, High, Advanced metastatic) guides adjuvant treatment. In 19 of 39 *p53abn-like NSMP* cases, the risk group would change (2 from Low to High, 7 from Intermediate to High, and 10 from High-intermediate to High). The remaining 20 cases (2 Intermediate risk and 18 High risk) would not have changed the risk group. Thus *p53abn-like NSMP* classification is potentially highly impactful on patient management, independent of other

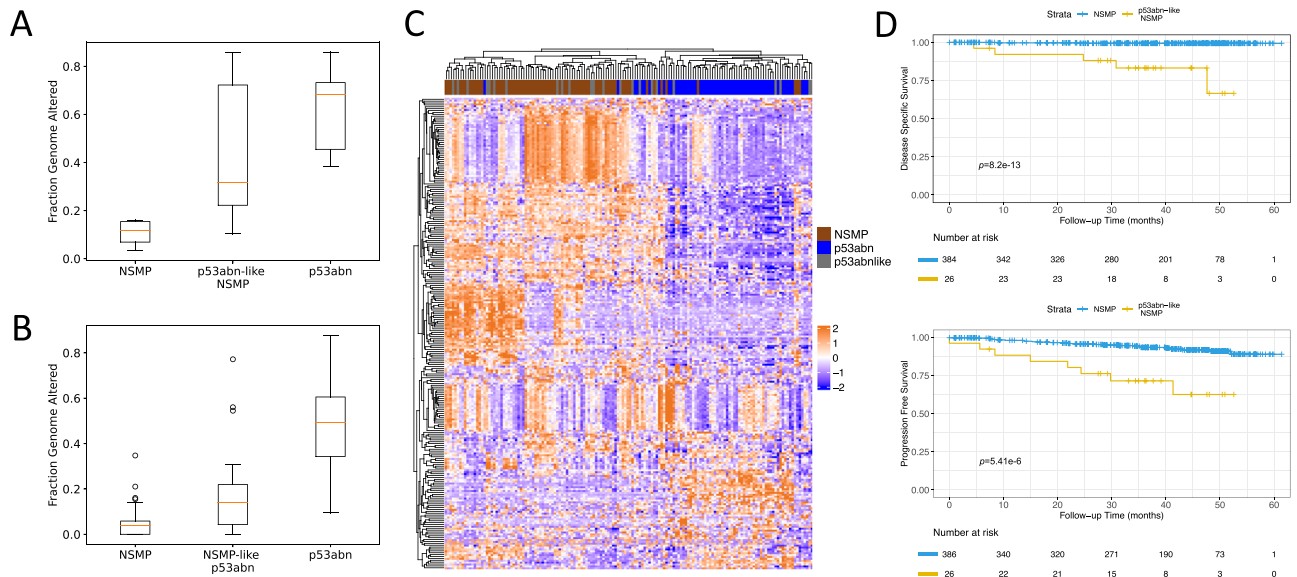

**Fig. 4 | Molecular profiling of *p53abn-like NSMP* cases.** Boxplots of copy number burden (i.e., fraction genome altered) in NSMP, *p53abn-like NSMP*, and p53abn cases in the (**A**) BC validation cohort (6 NSMP, *7 p53abn-like NSMP*, 5 p53abn) and (**B**) TCGA (69 NSMP, *21 p53abn-like NSMP*, 56 p53abn) cohorts. In box plots in A and B, the central line represents the median, while the bottom and top edges of the box correspond to the 25th and 75th percentiles, respectively. The whiskers extend to the most extreme data points that are not identified as outliers. Any data points beyond the lower and upper whiskers are considered outliers. **C** Gene expression profiles associated with the *p53abn-like NSMP* (n = 21), NSMP (n = 69), and p53abn (n = 56) tumors in the TCGA cohort. **D** KM curves associated with PFS and DSS (where available) in the CC validation set (Source data are provided as a Source Data file).

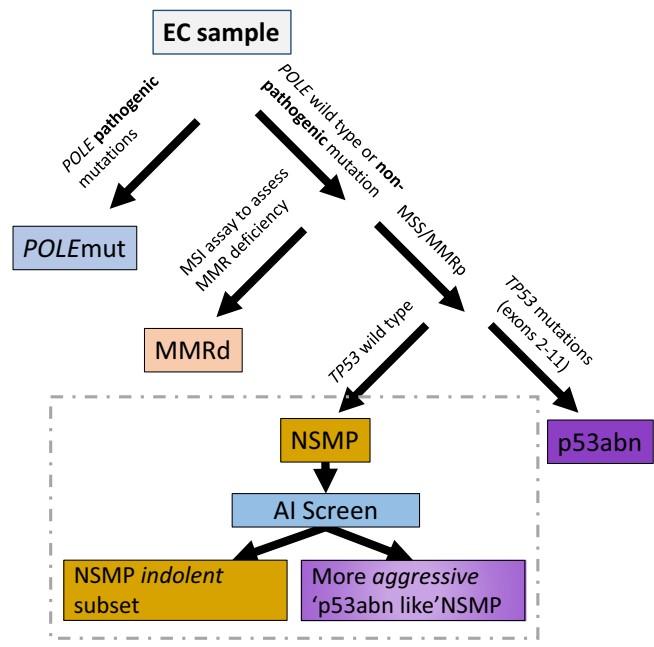

**Fig. 5 | AI-enhanced NSMP stratification.** AI-enhanced NSMP stratification. The refined classification scheme that leverages AI screening as a supplementary stratification mechanism within the NSMP molecular subtype.

clinical and pathological parameters such as stage, histotype, grade, and lymphovascular invasion.

## Discussion

Although many patients with endometrial carcinoma may be cured by surgery alone, about 1 in 5 patients have more aggressive disease and/or have the disease spread beyond the uterus at the time of diagnosis. Identifying these at-risk individuals remains a challenge, with current tools lacking precision. Molecular classification offers an objective and reproducible classification system that has strong prognostic value;

improving the ability to discriminate outcomes compared to conventional pathology-based risk stratification criteria. However, it has become apparent that within molecular subtypes and most profoundly within NSMP ECs, there are clinical outcome outliers. The current study addresses this diversity by employing AI-powered histopathology image analysis, in an attempt to identify clinical outcome outliers within the most common molecular subtype of endometrial cancer (Fig. 5). Our results have several clinical and biological implications.

To be clear, for some molecular subtypes, such as *POLE*mut endometrial cancers with almost uniformly favorable outcomes, no further stratification, at least within Stage I-II disease (encompassing > 90% of *POLE*mut ECs), is needed. Multiple studies, as well as meta-analyses[39], have shown that in patients with *POLE*mut endometrial cancers, additional pathological or molecular features are not associated with outcomes, i.e., are not prognostic, as *POLE* is the overriding feature that determines survival. However, for NSMP endometrial cancers, additional stratification tools are greatly needed. Designation of NSMP is the last step in molecular classification, only defined by what molecular features it does **not** have; that is without pathogenic *POLE* mutations, without mismatch repair deficiency or p53 abnormalities as assessed by IHC. This leaves a large group of pathologically and molecularly diverse tumors with markedly varied clinical outcomes.

Our AI-based histomorphological image analysis model identified a subset of NSMP endometrial cancers with inferior survival. This subset of patients encompassed approximately 20% of NSMP tumors which are the most common molecular subtype, representing half of endometrial cancers diagnosed in the general population, and thus account for 10% of all ECs. Our results suggest that clinicopathological, IHC, gene expression profiles, or NGS molecular markers (except for copy number burden to some extent) may not be able to identify these p53abn-like outliers. The AI classifier was able to identify those tumors with *TP53* mutations (but normal p53 immunostaining), a result we view as encouraging, in that these are "false negative" cases using the IHC classification and more appropriately assigned as p53abn, but even when these were removed from consideration AI imaging discerned other patients with NSMP EC where no molecular or pathological features would have identified them as having inferior outcomes.

Of note, our results corroborated with a recent report that identified a similar subset of NSMP cases with higher nuclear atypia in 3% of NSMP cases ($n = 4$ out of 120) with poor outcomes, although this difference was not statistically significant likely due to small sample size and differences in the image analysis models ($p = 0.13$)[23]. Taken together, AI applied to histomorphological images of routinely generated H&E slides appears to enable a more encompassing and easily implementable stratification of NSMP tumors and provides greater value than any single or combined pathological/molecular profile could achieve.

Molecular characterization of the identified subtype using sWGS suggests that these cases harbor an unstable genome with a higher fraction of altered genome, similar to the p53abn group but with a lesser degree of instability. These results suggest that the identified subgroup based on histopathology images is biologically distinct. Furthermore, our gene expression analysis revealed the upregulation of PI3k-Akt, Wnt, and Cadherin signaling pathways both in *p53abn-like NSMP* and p53abn groups (compared to NSMP). All these results suggest genomic and transcriptomic similarities between the *p53abn-like NSMP* and p53abn cases and potential defects in the DNA damage repair process as a possible biological mechanism. However, in spite of the fact that similar gene expression pathways were implicated in both groups and H&E images of both groups as assessed by AI had resemblances, expression data analysis revealed minimal overlap between the differentially expressed genes in both p53abn and *p53abn-like NSMP* ECs compared to other NSMP cases. This observation suggests that they may have different etiologies and warrants further biological interrogation of these groups in future studies.

Certainly, others have attempted to refine stratification within early-stage endometrial cancers, including within the molecularly defined NSMP subset. PORTEC4a used a combination of pathologic and molecular features (MMRd, L1CAM overexpression, *POLE*, *CTNNB1* status) to identify low, intermediate, and high-risk individuals assigned to favorable, intermediate, and unfavorable risk groups which then determined observations vs. treatment[40]. TAPER/EN.10 also stratifies early-stage NSMP tumors by pathological (e.g., histotype, grade, LVI status) and molecular features (*TP53*, ER status) to identify those individuals appropriate for de-escalated therapy[41]. In a retrospective series, key parameters of ER and grade have been suggested to discern outcomes within NSMP. ER status was also demonstrated to stratify outcomes in patients with NSMP ECs enrolled in clinical trials[42]. However, even in-depth profiling of apparent low-risk ECs has failed to find pathogenic features that would discern individuals who develop recurrence from other apparent indolent tumors[43]. Stasenko et al.[43]. assessed a series of 486 cases of 'ultra-low risk' endometrial cancers defined as stage 1 A with no myoinvasion, no LVI, grade 1 of which 2.9% developed recurrence with no identifiable associated clinical, pathological, or molecular features[43]. Current treatment guidelines, even where molecular features are incorporated, offer little in terms of directing management within NSMP endometrial cancers beyond consideration of pathological features, leaving clinicians to struggle with optimal management[12]. A more comprehensive stratification tool within NSMP endometrial cancers would be of tremendous value, and AI discernment from histopathological images as a tool that can be readily applied to H&E slides that are routinely generated as part of the practice is appealing.

Our proposed AI model also identified a subset of p53abn ECs with marginally superior DSS and resemblance to NSMP (*NSMP-like p53abn*) as assessed by H&E staining. Further investigation of the identified groups and deep molecular and omics characterization of this subset of p53abn ECs may in fact aid us in refining this subtype and identifying a subset of p53abn cases with statistically superior outcomes.

This study utilizes AI for refining endometrial cancer molecular subtypes. In general, such studies to generate new knowledge using AI in histopathology are extremely sparse as a majority of the effort has focused on recapitulating the existing body of knowledge (e.g., to diagnose cancer, to identify histological subtypes, to identify known molecular subtypes). This study moves beyond the mainstream AI applications within the current context of standard histopathology and molecular classification. This enables us to direct efforts to understand the biological mechanisms of this subset. This could present an exciting opportunity to utilize the power of AI to inform clinical trials and deep biological interrogation by adding more precision in patient stratification and selection.

AI histopathologic imaging-based application within NSMP enables discernment of outcomes within the largest endometrial cancer molecular subtype. It can be easily added to clinical algorithms after performing hysterectomy, identifying some patients (*p53abn-like NSMP*) as candidates for treatment analogous to what is given in p53abn tumors. Furthermore, the proposed AI model can be easier to implement in practice (for example, in a cloud-based environment where scanned routine H&E images could be uploaded to a platform for AI assessment), leading to a greater impact on patient management. Furthermore, we envision that an AI algorithm, after appropriate validation, could be utilized on diagnostic biopsy specimens, along with molecular subtype markers (p53, MMR, POLE). This would allow diagnosis of molecular subtype and further classify NSMP into lower risk and higher risk (p53abn-like), with the former patients being candidates for de-escalation of treatment (e.g., simple hysterectomy in the community) and the latter group potentially directed to cancer centers for lymph node assessment, omental sampling and directed biopsies given a higher likelihood of upstaging. It is possible that with further refinement and validation of the algorithm, which can be run in minutes on the diagnostic slide image, that it could take the place of molecular subtype markers, saving time and money.

## Methods

### Ethics statement

The Declaration of Helsinki and the International Ethical Guidelines for Biomedical Research Involving Human Subjects were strictly adhered to throughout this study. All protocols for this study, including the waiver of consent, have been approved by the University of British Columbia/BC Cancer Research Ethics Board. Participants did not receive compensation.

### Histopathology slide digitization

Histopathology slide images associated with the TCGA cohort were acquired from the TCGA GDC portal (https://portal.gdc.cancer.gov). Histopathology slides associated with the Canadian cohorts as well as the Tübingen University Women's Hospital were scanned using an Aperio AT2 scanner.

### AI tumor-normal classifier and automatic annotation

The downstream tumor subtype classifier relies on the tumor areas of the tissues. Given that the manual annotation of all slides by pathologists is tedious and time-consuming, we first trained a deep learning model to identify the tumor areas of the slides automatically (Supplementary Fig. 7). To train the model, we utilized 27 slides that were annotated by a board-certified pathologist. First, we split the slides into training (51.8%), validation (22.2%), and testing (26%) sets. To identify the tumor regions of WSIs, we divided them into smaller tiles referred to as patches and extracted 5091 (2167 tumor, 2924 stroma) non-overlapping patches. A maximum of 200 patches with a size of $512 \times 512$ pixels at 20x objective magnification were extracted from the annotated regions of each slide. As the baseline architecture for our classifier, we exploited ResNet18[44], a simple and effective residual network, with the pre-trained ImageNet[45] weights. We trained the model with the learning rate and weight decay of 1e-4 for five epochs using the Adam optimizer[46]. As the amount of tumor and stroma patches were not equal, we used a balanced sampler with a batch size

## BOX 1
# Subtype prediction algorithm

Input: $P_i = \left\{ p_{i1}, .., p_{ik_i} \right\}$: a set of extracted patches from the $i^{th}$ image, i.e., $I_i$

1: for $j \leftarrow 1$ to $k_i$ do
2: $z_j \leftarrow \mathcal{F}_{Conv2D}(p_{ij})$
3: end for
4: Let $W \in \mathbb{R}^q$ and $V \in \mathbb{R}^{q \times d}$ be the attention parameters
5: for $j \leftarrow 1$ to $k_i$ do
6: $a_j \leftarrow \exp(W^\top \tanh(Vz_j)) / \sum_{t=1}^{K_i} \exp(W^\top \tanh(Vz_t))$
7: end for
8: $\bar{z} \leftarrow \sum_{j=1}^{k_i} a_j z_j$
9: $z_\sigma \leftarrow \frac{k_i}{k_i-1} \sum_{j=1}^{k_i} a_j \left(z_j - \bar{z}\right)^2$
10: $z \leftarrow \bar{z} \oplus z_\sigma$
11: $Y_i \leftarrow \mathcal{F}_{FC}(z)$
Output: $Y_i$: the predicted subtype of $I_i$

of 150 which meant that in each batch, the model was trained using 75 tumor patches and 75 stroma patches. The resulting classifier achieved 99.76% balanced accuracy on the testing set, indicating the outstanding performance of this tumor/non-tumor model (Supplementary Table 5). The trained model was then applied to detect tumor regions on the rest of the WSIs. To that end, we extracted patches with identical size and magnification to the training phase. To achieve smoother boundaries for the predicted tumor areas we enforced a 60% overlap between neighboring patches. In addition, to reduce false positives we used a minimum threshold probability of 90% for tumor patches. Finally, for consistency, we applied the trained model on the discovery set, including the cases that were manually annotated by a pathologist.

### Deep learning models for tumor subtype classification

Due to the lack of pixel-wise annotations, we employed variability-aware multiple instance learning (VarMIL)[33] that utilizes the multiple instance learning technique in which an image is modeled as an instance containing a bag of unlabeled patches or tiles. The algorithm in Box 1 elaborates on the prediction mechanism of $\mathcal{F}_{FC}$ in detail. $z_j \in \mathbb{R}^d$ consists of three sections: a feature extractor network (d), attention layers, and classification layers ($a_j \in \mathbb{R}$). First, the feature extractor network computes feature embeddings ($z \in \mathbb{R}^{2d}$) for the extracted patches of an instance (i.e., image), where $z_\sigma \in \mathbb{R}^d$ is the dimension of the embeddings. Second, given that patches of a given image are not necessarily equally important in subtype prediction, an attention mechanism calculates the contribution of each patch ($\bar{z} \in \mathbb{R}^d$) based on its embedding. Subsequently, VarMIL computes the image's representation ($z \in \mathbb{R}^{2d}$) into account alongside their weighted average ($z_\sigma \in \mathbb{R}^d$)). Finally, the model feeds the derived representation as the input of the classification section to predict the subtype. To avoid over-fitting, we employed a variety of augmentation methods including horizontal and vertical flipping, color jitter, size jitter, random rotation, and Cutout[47]. Furthermore, we utilized early stopping[48] as an additional form of regularization in training, and if the validation loss did not decrease after five epochs, we decreased the learning rate. Furthermore, we stopped the training if the validation loss did not decrease after 10 consecutive epochs. We devised a two-step training procedure for the proposed network, in which the feature extractor network was trained independently from the attention and classification layers. First, we trained the feature extractor, ResNet34[44] (d = 512). For the attention and classification layers, we selected a

multilayer perceptron (MLP) with a single hidden layer with 128 nodes (q = 128). We trained these layers with the same number of epochs and weight decay as before but with a learning rate of 1e-5. Models were trained using a single dgxV100 GPU with 32GB RAM. The programming language was PyTorch[49] (version 1.8.0), and we selected the hyperparameters experimentally.

We further assessed the robustness of our findings with five other models formulated on distinct concepts: (1) Vanilla[37,50], (2) Histogram-Based[51], (3) Iterative Draw and Rank Sampling (IDaRS)[52], (4) Attention-based[53,54], and (5) Vector of Locally Aggregated Descriptor (VLAD)[55,56], CLAM-MB, CLAM-SB, and TransMIL.

*(1) Vanilla* is a simple and frequently used concept in digital pathology[37,57]. In this setting, we train a DL model on the extracted patches from a histopathology slide in a fully supervised manner. Here, each patch's label corresponds to the subtype of its corresponding histopathology slide. The process involves passing patches through convolutional layers and feeding the generated feature maps into fully connected layers. The model is trained using the cross-entropy loss function[58], similar to standard classification tasks.

*(2) IDaRS* shares similar assumptions with Vanilla, involving training a model on image patches in a fully supervised manner and assigning the image's label to its patches[52]. However, unlike Vanilla, where all extracted patches are used in training, IDaRS employs a selection procedure. Only informative patches that contribute to the image's subtype are included during training. The selection algorithm utilizes the Monte-Carlo[59] sampling approach.

*(3) The Histogram-Based* concept[51] addresses the task of identifying a slide's subtype, similar to IDaRS and Vanilla, by transforming a weakly supervised problem into a fully supervised one. A key distinction of this concept is the integration of a histogram and a classification module, instead of relying on majority voting. This modification improves the model's interpretability without significantly increasing the parameter count.

*(4) DeepMIL*[60] combines the concepts of MIL and attention. It leverages MIL techniques, treating an image as a collection(bag) of unlabeled patches, while the attention-based approach maintains the nature of the weakly supervised task, in contrast to the previously mentioned concepts. This perspective removes the need to assign labels to individual patches within an image. Moreover, it recognizes that patches within an image have varying degrees of importance to its subtype, and their contributions are calculated using an attention mechanism.

*(5) VLAD*[55], a family of algorithms, considers histopathology images as Bag of Words (BoWs), where extracted patches serve as the words. Due to its favorable performance in large-scale databases, surpassing other BoWs methods, we adopt VLAD as a technique to construct slide representation[55].

*(6) CLAM*[61] adopts an attention-based pooling function to aggregate patch-level features to form slide-level representations for classification. By ranking all patches within a slide, the model assigns attention scores to each patch, revealing their unique contributions and significance to the overall slide-level representation for a specific class. In addition, CLAM utilizes instance-level clustering over identified representative regions to constrain and refine the feature space. Distinguishingly, CLAM-SB utilizes a single attention branch for aggregating patch information, while CLAM-MB employs multiple attention branches, corresponding to the number of classes used for classification.

*(7) TransMIL*[62] represents a transformer-based methodology devised for the classification of whole slide histopathology images. This framework incorporates both morphological and spatial information through a comprehensive consideration of contextual details surrounding a singular area and the inter-correlation between distinct areas. Furthermore, the approach employs a Pyramid Position Encoding Generator (PPEG) module for transforming local features and encoding positional information.

## Identification of *p53abn-like NSMP*s

The initial hypothesis was that NSMP cases with a poor prognosis resemble p53abn morphologically. Assuming the hypothesis is correct, subtype classifiers should label cases in this group as p53abn. Using the same rationale, we partitioned the NSMP subtype into two subgroups: *p53abn-like NSMP* and the remaining NSMP cases. To this end, we devised a voting system based on the classifiers' consensus. If the fraction of classifiers predicting an NSMP case as p53abn exceeded a specified confidence threshold, the image was labeled as *p53abn-like NSMP*; otherwise, the image was labeled as NSMP. In this work, we labeled a sample as *p53abn-like NSMP* when an NSMP sample, based on ProMisE, was classified as p53abn in more than seven out of the 10 cross-validation classifiers.

## Unsupervised clustering of NSMP patch representations

To investigate the robustness of our results in identifying *p53abn-like NSMPs* and visualize the distribution of the patch representations, we employed a two-step approach. In the first step, we applied KimiaNet[34] to the patches that were extracted from the histopathology slides associated with the NSMP EC cases. KimiaNet is a deep model trained on a large set of histopathology data, to encode each patch with dimensions of $512 \times 512$ pixels into a compact $1024 \times 1$ vector. By leveraging the embeddings from KimiaNet's last pooling layer, we condensed the essential features of each patch into a representative vector. In the second step, we applied Uniform Manifold Approximation and Projection (UMAP)[63], a dimensionality reduction technique, to project the encoded vectors of all the patches within the NSMP and *p53abn-like NSMP* onto a two-dimensional space. UMAP excels at preserving both local and global structures of high-dimensional data, enabling us to visualize the relationships and patterns within the encoded patches in a more interpretable manner.

## Targeted point mutation profiling

Targeted mutation profiling was performed as part of our team's previous efforts[4,5,64,65]. Briefly, the exon capture libraries were sequenced using the Illumina Genome Analyzer (GAIIx), MiSeq, or Sanger sequencing. The reads were aligned to the human genome using the BWA aligner version v0.5.9. and SNVs were called by a combination of binomial exact test and MutationSeq as previously described[66,67]. To remove the germline mutations, the predicted SNVs were filtered through dbSNP, 1000 Genome (http://www.1000genomes.org/), and the control normals. All SNVs were profiled by MutationAssessor[68] for the functional impact of the missense mutations. snpEff (http://snpeff.sourceforge.net/) was used to find splice site mutations. All silent mutations were removed. The indels were filtered by the control normals and then profiled by Oncotator (http://www.broadinstitute.org/oncotator/).

## Survival analysis

We assessed the significance of subgroups using the Kaplan-Meier (KM) estimator on two survival endpoints: Disease Specific Survival (DSS) and Progression Free Survival (PFS). Survival outcomes were not accessible for four (1.47%; three NSMP and one *p53abn-like NSMP*) and two (1.03%; one NSMP and one *p53abn-like NSMP*) patients in the discovery and validation sets, respectively. In some individuals, clinical data were partially available (for example, survival data of a patient only contained DSS while PFS was unknown), explaining why the number of cases varies among KM curves for the same set. In addition, given that the TCGA survival data lacked DSS, the German cohort served as the discovery set for the DSS KM curves.

## Shallow whole genome sequencing (cohort and experiments)

DNA was extracted (GeneRead FFPE DNA kit from Qiagen) from FFPE core tumor samples and was sheared to 200 bp using a Covaris S220. Libraries were constructed using the ThruPlex DNA-seq kit (Takara) with seven cycles of amplification (library prep strategy from Brenton Lab similar to the one published in 2018)[69]. Library quality was assessed using the Agilent High Sensitivity DNA kit (Agilent Technologies), and pooled libraries were run on the Illumina NovaSeq at the Michael Smith Genome Sciences Center targeting 600 M reads per pooled batch. The sWGS data was run through basic processing which includes trimming with Trimmomatic[70], alignment with bwa-mem2[71], duplicate removal with Picard[72], and sorting with samtools[73]. Sequencing coverage and quality were evaluated using fastQC[74] and samtools. If acceptable, the data was passed along to the next step of determining genomic copy numbers (QDNAseq[75] + rascal[76]) and signature calls. The signature calling step uses techniques including mixture modeling and non-negative matrix factorization and is composed mostly of software from the CN-Signatures[69] package with a few in-house modifications and additions. Interim data munging and ETL (extract, transform, load) are done primarily in bash and R (tidyverse), while visualization and plotting is performed mostly just in R using ggplot2 and pheatmap.

## Gene expression analysis

For expression profiling, we used RNA-seq profiles obtained from the TCGA-UCEC cohort[3]. Specifically, we used the GDC data portal[77] to download primary tumors sequenced on the Illumina Genome Analyzer platform with patient IDs matching those used in our study. Raw, un-normalized counts were used. DeSeq2[78] was used to process the raw count matrix and perform differential expression analysis (DEA) and hierarchical clustering. Samples were categorized as NSMP, *p53abn-like NSMP*, and p53abn. Genes with a total count of five or less were removed. Counts were normalized using DeSeq2's variance-stabilizing transform tool. The 500 most variable genes based on DEA were kept for hierarchical clustering. Per-gene Z-scaling was applied to normalize the clustering features. Finally, the complete-linkage method was used for both gene-clustering and sample-clustering. Subsequent pathway analysis on the list of differentially expressed genes was performed using the Reactome[79] FI plugin in Cytoscape[80].

## Statistical assessment

A two-sided log-rank test was utilized to assess the significance of the difference between KM curves for the identified patient groups. In

addition, the significance of groups for the enrichment of specific genomic or molecular features was assessed using the Fisher's exact test and the Mann-Whitney U rank test for discrete and continuous data, respectively. Throughout all experiments, $p < 0.05$ was regarded as the significance level.

## Reporting summary

Further information on research design is available in the Nature Portfolio Reporting Summary linked to this article.

## Data availability

All histopathology slide images in this study can be obtained by direct email to the corresponding author. All data access is subject to institutional permission and compliance with ethics from the corresponding institutions. Data can only be shared for non-commercial academic purposes and will require a data user agreement. The whole-genome sequencing data for this study have been deposited in the European Nucleotide Archive (ENA) at EMBL-EBI under accession number PRJEB60600. Endometrial (uterine) carcinoma samples from The Cancer Genome Atlas (TCGA), used in this study, can be freely downloaded from [https://portal.gdc.cancer.gov/analysis_page?app=Projects]. The exome-wide and targeted point mutation data discussed in this manuscript were previously published in earlier studies[4,5,64,65]. The aggregated mutation calls including the genomic coordinates of the mutations and reference and tumor alleles were extracted directly from those studies[4,5,64,65] and can be found in the Source Data of Fig. 3B. Source data are provided with this paper.

## Code availability

The code used in this manuscript is available on https://github.com/AIMLab-UBC/EC-p53abnlike-AIclassifier

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

## Acknowledgements

This work was supported by Terry Fox Research Institute (J.M., A.B., D.G.H., grant number: 1116), Canadian Institute of Health Research (A.B., grant number: 201903PJT-418734), Natural Sciences and Engineering Research Council of Canada (A.B., grant number: RGPIN-2019-04896), Michael Smith Foundation for Health Research (A.B., grant number: SCH-2021-1546), Canada Research Chair (J.M., S.J.M.J., D.G.H), Canada Foundation for Innovation/BC Knowledge Development Funds (A.B., grant number: 41144), OVCARE Carraresi and VGH UBC Hospital Foundation (A.B.). The funders had no involvement in study conception, data collection, data analysis, data interpretation, writing of the report, or publication decisions.

## Author contributions

A.D. and H.F. were the research project leaders and led and designed all data analysis. M.A. performed the data analysis in the revision of the manuscript. A.D. implemented all the deep learning pipelines. M.W. and A.B. performed the gene expression analysis. M.A. and A.K. performed image analysis. D.C. and A.J. contributed to the clinical review and case selection for molecular profiling. D.F. performed the histopathology slide annotations. P.Ah. contributed to software infrastructure for slide annotation. M.D. and H.F. performed bioinformatics analysis. P.Ab and S.J.M.J. provided advice on machine learning analysis and provided computational resources. A.T. and S.L. contributed to clinical informatics and biobanking. C.B.G. and N.S. reviewed all specimens for histological and molecular pathology and contributed to manuscript writing. S.K. were responsible for the specimen and clinical data from Tübingen University. J.N.M. and C.B.G. contributed to cohort construction, tumor banking, and the initial draft of the manuscript. A.D., H.F., and A.B. wrote the first draft of the manuscript. D.G.H., N.S., and J.N.M. provided oversight, edited the manuscript, and supervised the study. A.B. conceived and oversaw the project and is the senior corresponding author. All authors have reviewed and approved the manuscript content.

## Competing interests

The authors declare no competing interests.
