## [Peer Review File · Nature Communications]

AI-based histopathology image analysis reveals a distinct subset of endometrial cancersREVIEWER COMMENTS

Reviewer #1 (Remarks to the Author): Expert in endometrial cancer genomics, clinical research, and histopathology

- What are the noteworthy results?

This paper constructed an AI-based histopathology image analysis system to differentiate between NSMP and p53abn endometrial subtypes. Using this AI-base system, the author consequently identified a sub-group of NSMP EC (p53abn-like NSMP) with inferior prognosis comparing with other NSMP patients. The author believed their AI-based image analysis system has the power to detect prognostically different and otherwise unrecognizable subsets of EC.

The noteworthy result of this work is that the AI-based histopathology image analysis system can identify the p53abn-like NSMP with inferior prognosis than other NSMP patients.

- Will the work be of significance to the field and related fields? How does it compare to the established literature? If the work is not original, please provide relevant references.

The application of AI-based image analysis system in differentiating molecular classifications has been reported in recent years. The new information provided by this paper is that the AI system can further identify a sub-group of p53abn-like NSMP patients with poorer prognosis which might help clinical decisions for adjuvant treatment after surgery. However, as we all know that NSMP is a group of patients with different prognosis. Molecular classification is not the only known risk factors predicting the prognosis of the patients. Clinical doctors combine all the information, including tumor grade, LVSI, stage, as well as IHC markers, etc. to evaluate the prognosis of the patient and make clinical decisions. It is not known whether the AI-system in this paper has superior predictive power than the traditional risk classification system to identify NSMP patients with poorer prognosis. The author should provide consolidate evidence to demonstrate the advantage of their AI-system comparing with the present risk classification system.

- Does the work support the conclusions and claims, or is additional evidence needed?

The author compared the prognosis of different patient groups identified by their AI-system. However, as the author mentioned in the paper, the follow-up data of some patients were missed due to various reasons. How many of the patients' data were missed and how did

the author deal with these missing data? Because the patient number in the study is not large, the missing data might affect the final results and consequently the conclusion drawn from the results.

- Are there any flaws in the data analysis, interpretation and conclusions? Do these prohibit publication or require revision?

The author did not mention what method do they use for molecular classification in the validation cohort. I assume they used ProMisE system. However, the concordance rate of copy number high defined by TCGA and abnormal p53 IHC used by ProMisE is about 90%. And the concordance rate of TP53 mutation and abnormal p53 IHC is about 90%. Which means that the discovery cohort defined by the TCGA has about 10% discrepancy with the validation cohort defined by ProMisE. It is possible that the p53abn-like NSMP EC are the real cases with CNH or TP53 mutation missed by p53 IHC. The author also showed by their sWGS that the p53abn-like NSMP had higher copy number variation than other NSMP cases. What about TP53 gene status in this group of patients? The author might explain clearly whether the advantage of their AI-system is to identify the real CNH patients missed by ProMisE classification system.

The author mentioned that Fig.3A shows an enrichment of estrogen receptor (ER) and PR positive cases in the p53abn-like NSMP. Is it enrichment of ER PR negative cases in p53abn-like NSMP comparing with other NSMP?

The author failed to find concordant results between discovery cohort and validation cohort after targeted sequencing of exonic regions. How to explain this or what is the implication of these findings?

- Is the methodology sound? Does the work meet the expected standards in your field?

Please provide the methodology for molecular classification in the validation cohort.

Did the author used hysterectomy sample only or endometrial biopsy sample?

What is the meaning of stromal part? Does this mean myometrial part?

- Is there enough detail provided in the methods for the work to be reproduced?

Big variations might exist in the EC lesion because of the heterogeneity of the tumor. What kind of protocol doses the author use to select the most representative slide for AI analysis?

Reviewer #3 (Remarks to the Author): Expert in machine learning, deep learning, and digital pathology in cancer

This paper presents a novel and valuable approach to endometrial cancer (EC) classification using artificial intelligence (AI)-powered histopathology image analysis. The authors successfully differentiated between p53abn and NSMP EC subtypes and identified a previously unrecognized sub-group of NSMP EC patients with inferior outcomes termed 'p53abn-like NSMP.' The study also explores the potential of using AI to refine the molecular subtypes of EC, which is an exciting development in the field of computational pathology. However, the paper has several notable shortcomings that need to be addressed before it can be considered for publication. These limitations are crucial to the study's scientific rigor, reproducibility, and overall impact:

1. Detailed information about the discovery cohort (368 patients) and the validation cohort (290 patients) should be provided, including patient demographics, clinical characteristics, and treatment regimens. Moreover, it is essential to clarify the criteria for selecting patients with p53abn-like NSMP EC and NSMP-like p53abn EC for the analysis. Explain how the cohorts were representative of the general EC population to ensure the study's generalizability.

2. The AI-based model achieved high performance in differentiating p53abn and NSMP ECs in the discovery and validation sets. However, were any external datasets or independent centers used for further validation to assess the model's generalization to different populations and staining protocols? If not, please discuss the potential limitations of the model's applicability in other clinical settings.

3. Although the study identified a novel subset of NSMP ECs with inferior survival termed 'p53abn-like NSMP,' it remains unclear how these findings can be translated into clinical practice. How do you envision implementing the AI model in routine clinical settings? What potential challenges and limitations might arise when integrating this approach into current diagnostic procedures and treatment decision-making?

4. The study employed various deep learning-based models for tumor subtype classification,

including Vanilla, Histogram-Based, IDaRS, Attention-based, and VLAD. However, how was the selection made for using these specific models? Were any other state-of-the-art models considered for comparison, and what were the reasons for excluding them?

5. In the unsupervised clustering of NSMP patch representations using KimiaNet and UMAP, how sensitive are the results to the choice of dimensionality reduction technique and the number of clusters? Did you conduct any sensitivity analyses to verify the stability of the clustering results?

6. The use of histopathology slides for training and validation raises questions about potential sources of bias and the generalizability of the findings. How do you address the potential heterogeneity in slide staining and imaging quality across different centers, which might impact the model's performance and applicability in other settings?

7. The study validated the AI model's performance on an external validation set from a single center (Vancouver cohort). Considering the variability in tissue processing and data collection across centers, were any efforts made to include additional external validation cohorts from different geographic locations to enhance the model's robustness and generalization?

8. The study identifies a subset of NSMP endometrial cancers with unique clinical outcomes. Can the authors propose plausible biological mechanisms underlying the observed differences in survival outcomes between the p53abn-like NSMP and other NSMP cases? Additional biological investigations may strengthen the study's implications.

RESPONSE TO REFEREES

REVIEWER 1 (R1) COMMENTS

R1 Comment 1: This paper constructed an AI-based histopathology image analysis system to differentiate between NSMP and p53abn endometrial subtypes. Using this AI-base system, the author consequently identified a sub-group of NSMP EC (p53abn-like NSMP) with inferior prognosis comparing with other NSMP patients. The author believed their AI-based image analysis system has the power to detect prognostically different and otherwise unrecognizable subsets of EC.

The noteworthy result of this work is that the AI-based histopathology image analysis system can identify the p53abn-like NSMP with inferior prognosis than other NSMP patients.

Response: We would like to thank the reviewer for recognizing the noteworthiness of AI in identifying a previously undescribed subset of NSMP ECs with inferior survival.

R1 Comment 2: The application of AI-based image analysis system in differentiating molecular classifications has been reported in recent years. The new information provided by this paper is that the AI system can further identify a sub-group of p53abn-like NSMP patients with poorer prognosis which might help clinical decisions for adjuvant treatment after surgery. However, as we all know that NSMP is a group of patients with different prognosis. Molecular classification is not the only known risk factors predicting the prognosis of the patients. Clinical doctors combine all the information, including tumor grade, LVSI, stage, as well as IHC markers, etc. to evaluate the prognosis of the patient and make clinical decisions. It is not known whether the AI-system in this paper has superior predictive power than the traditional risk classification system to identify NSMP patients with poorer prognosis. The author should provide consolidate evidence to demonstrate the advantage of their AI-system comparing with the present risk classification system.

Response: We thank the reviewer for their insightful comment. We agree that it is important to assess whether the finding of p53abn-like would be potentially impactful on patient care, independent of the many other factors used by clinicians to guide patient management. We therefore assessed whether the ESGO/ESTRO/ECP 2021 risk group (a widely used guideline for assignment of risk group based on clinical and molecular parameters, and notable for being the first to incorporate molecular subtype in risk group category). We found that in 19 of 39 *p53abn-like NSMP* the risk group would have changed to a higher risk group. We do not have sufficient numbers of cases to compare the outcomes of *p53abn-like NSMP* to other tumors within each of the risk groups, so did not attempt that analysis. The following paragraph was added as the final paragraph of the Results:

“Impact of p53abn-like NSMP on Risk Group assignment. We next sought to determine whether the finding of p53abn-like status by AI, in an NSMP endometrial carcinoma, would potentially change the risk group category i.e. if the tumor was classified as if it were p53abn molecular subtype rather than NSMP, would that impact on the final risk group assignment. The ESGO/ESTRO/ECP 2021 risk group classification is based on molecular subtype, stage, histotype, grade, lymphovascular invasion and the presence of residual disease (REF); this risk group (Low, Intermediate, High-intermediate, High, Advanced metastatic) guides adjuvant treatment. In 19 of 39 p53abn-like NSMP cases the risk group would change (2 from Low to High, 7 from Intermediate to High, and 10 from High-intermediate to High). The remaining 20 cases (2 Intermediate risk and 18 High risk) would not have changed risk group. Thus p53abn-like NSMP classification is potentially highly impactful on patient management, independent of other clinical and pathological parameters such as stage, histotype, grade and lymphovascular invasion.”

R1 Comment 3: The author compared the prognosis of different patient groups identified by their AI-system. However, as the author mentioned in the paper, the follow-up data of some patients were missed due to various reasons. How many of the patients' data were missed and how did the author deal with these missing data? Because the patient number in the study is not large, the missing data might affect the final results and consequently the conclusion drawn from the results.

Response: We have amended the manuscript and added a table (Supplemental Table 2) listing the number of patients with various follow-up data. More importantly, we have confirmed our findings on a new cohort of patients (n = 614) that have been collected from 26 hospitals across Canada (Main text section "*Validation of p53abn-like NSMP subtype in a multi-centre dataset*" in the Results). Having confirmed the findings on two separate cohorts from a variety of institutions emphasizes the validity of our findings. To our knowledge, our combined datasets are the largest collection of p53abn and NSMP molecular subtypes of EC yet published (2,318 slides, 1,272 patients).

R1 Comment 4: The author did not mention what method do they use for molecular classification in the validation cohort. I assume they used ProMisE system. However, the concordance rate of copy number high defined by TCGA and abnormal p53 IHC used by ProMisE is about 90%. And the concordance rate of TP53 mutation and abnormal p53 IHC is about 90%. Which means that the discovery cohort defined by the TCGA has about 10% discrepancy with the validation cohort defined by ProMisE. It is possible that the p53abn-like NSMP EC are the real cases with CNH or TP53 mutation missed by p53 IHC. The author also showed by their sWGS that the p53abn-like NSMP had higher copy number variation than other NSMP cases. What about TP53 gene status in this group of patients? The author might explain clearly whether the advantage of their AI-system is to identify the real CNH patients missed by ProMisE classification system.

Response: We thank the reviewer for their great comment. We performed targeted sequencing of exonic regions in TP53 gene which revealed an enrichment of p53abn-like NSMP cases with TP53 mutations (Fisher's exact test p-values = 3.14×10^{-4}). More specifically, we identified eight (out of 39) p53abn-like NSMP tumors that had normal p53 IHC results (hence classified as NSMP by ProMisE classifier) but in fact harbored TP53 mutations by sequencing. These cases are examples of the well-known phenomenon of normal p53 protein levels despite there being a pathogenic mutation, which occurs in <5% of cases. However, even after removing these eight TP53 mutant cases, the worse prognosis of p53abn-like NSMP tumors persisted (Fig. 3B). Our ML model, therefore, identifies tumors with false negative immunostaining for p53, i.e., they lack mutant pattern protein expression despite having a TP53 mutation, but also identifies a subset of NSMP cases with features of p53abn morphology by H&E but neither mutation pattern immunostaining nor a mutation in sequencing TP53, and these have inferior survival compared to the rest of the NSMP cases.

We have amended the text under section "Genomic characterization of p53abn-like NSMP cases" to report these findings.

R1 Comment 5: The author mentioned that Fig.3A shows an enrichment of estrogen receptor (ER) and PR positive cases in the p53abn-like NSMP. Is it enrichment of ER PR negative cases in p53abn-like NSMP comparing with other NSMP?

Response: Thank you for this comment. We meant enrichment for ER/PR positive cases in the p53abn-like NSMP cases compared to the NSMP cases that were classified as NSMP by AI. We have amended the text to clarify this finding accordingly (section "*Comparison of NSMP and p53abn-like NSMP*").

R1 Comment 6: The author failed to find concordant results between discovery cohort and validation cohort after targeted sequencing of exonic regions. How to explain this or what is the implication of these findings?

Response: This is an excellent point. We believe this observation is due to multiple hypothesis testing and the fact that we had not corrected for multiple test when assessing enrichment for single gene mutations. After correcting for multiple hypothesis testing, we can report that there is no single gene mutation that is enriched in either p53abn-like NSMPs and NSMPs classified by AI and the results are concordant between discovery cohort and validation cohort after targeted sequencing of exonic regions. Given that our analysis was repeated on multiple genes, we realized that we had to perform multiple hypothesis testing after statistical significance assessment and therefore, confirm that the only gene that was reported to be enriched in p53abn-like NSMPs in the validation cohort (i.e., CTNNB1) is not significant anymore after correction for multiple hypothesis testing. We have now amended the manuscript to correct our statement.

R1 Comment 7: Please provide the methodology for molecular classification in the validation cohort. Did the author use hysterectomy sample only or endometrial biopsy sample? What is the meaning of stromal part? Does this mean myometrial part?

Response: The ProMisE classifier was used in all three cohorts. We have utilized hysterectomy material in all cohorts of this study. With respect to the question related to stroma within the context of annotating slides to detect tumor regions, we had meant tissue without tumor cells, including stroma. As such, to avoid confusion, we have reworded the sentence from:

“A subset of 27 whole section H&E slides from the TCGA cohort were annotated by a board-certified pathologist (DF) using a custom in-house histopathology slide viewer (cPathPortal) to identify areas containing tumor and stromal cells.”

to

“A subset of 27 whole section H&E slides from the TCGA cohort were annotated by a board-certified pathologist (DF) using a custom in-house histopathology slide viewer (cPathPortal) to identify areas containing tumor and non-tumor cells (myometrium, endometrial stroma, and benign endometrial epithelium.”

R1 Comment 8: Big variations might exist in the EC lesion because of the heterogeneity of the tumor. What kind of protocol does the author use to select the most representative slide for AI analysis?

Response: Our AI analysis workflow has been deployed on TCGA and archival slides from Vancouver, laboratories across Canada (in the new validation cohort now added to the study) and Germany. We do not exclude any slides from AI analysis and instead use all the available slides for final subtyping. While there can be morphological heterogeneity within EC, the molecular subtype is uniform throughout in almost all cases, as shown in studies done by us and others correlating molecular subtype as determined based on biopsy and hysterectomy specimens. Therefore, the only criterion used for selection of a representative slide of tumor when these cases were originally collected was choosing a section with sufficient well-fixed tumor to perform molecular analysis, which, we believe, should not introduce a systematic bias.

REVIEWER 3 (R3) COMMENTS

R3 Comment 1: This paper presents a novel and valuable approach to endometrial cancer (EC) classification using artificial intelligence (AI)-powered histopathology image analysis. The authors successfully differentiated between p53abn and NSMP EC subtypes and identified a previously unrecognized sub-group of NSMP EC patients with inferior outcomes termed 'p53abn-like NSMP.' The study also explores the potential of using AI to refine the molecular subtypes of EC, which is an exciting development in the field of computational pathology. However, the paper has several notable shortcomings that need to be addressed before it can be considered for publication. These limitations are crucial to the study's scientific rigor, reproducibility, and overall impact.

Response: We thank the reviewer for their encouraging evaluation of our work. We have carefully considered the comments and believe that we have addressed them in full, as outlined below.

R3 Comment 2: Detailed information about the discovery cohort (368 patients) and the validation cohort (290 patients) should be provided, including patient demographics, clinical characteristics, and treatment regimens. Moreover, it is essential to clarify the criteria for selecting patients with p53abn-like NSMP EC and NSMP-like p53abn EC for the analysis. Explain how the cohorts were representative of the general EC population to ensure the study's generalizability.

Response: As requested, we have provided patient demographics, clinical characteristics and treatment regimens for the cohorts in Tables 1-3 and Supplementary Tables 2-3. Furthermore, to demonstrate the generalizability of the models, we have added a new dataset (n = 614 patients) that have been collected from 26 hospitals across Canada (Main text section "*Validation of p53abn-like NSMP subtype in a multi-centre dataset*" in the Results). Having confirmed the findings on two separate cohorts from a range of institutions (e.g. community hospitals and cancer centers) emphasizes the validity of our findings. To our knowledge, our combined datasets are the largest collection of the two molecular subtypes of EC.

R3 Comment 3: The AI-based model achieved high performance in differentiating p53abn and NSMP ECs in the discovery and validation sets. However, were any external datasets or independent centers used for further validation to assess the model's generalization to different populations and staining protocols? If not, please discuss the potential limitations of the model's applicability in other clinical settings.

Response: We are happy to report that we have confirmed our findings on a new cohort (n = 614 patients) that was collected from various hospitals across Canada (Main text section "*Validation of p53abn-like NSMP subtype in a multi-centre dataset*" in the Results).

R3 Comment 4: Although the study identified a novel subset of NSMP ECs with inferior survival termed 'p53abn-like NSMP,' it remains unclear how these findings can be translated into clinical practice. How do you envision implementing the AI model in routine clinical settings? What potential challenges and limitations might arise when integrating this approach into current diagnostic procedures and treatment decision-making?

Response: This is a great point. The last paragraph of the Discussion touches on the potential clinical utility of our finding and how this could be implemented in the clinic. For convenience, we have included this paragraph below:

"AI histopathologic imaging-based application within NSMP enables discernment of outcomes within the largest endometrial cancer molecular subtype. It can be easily added to clinical algorithms after performing hysterectomy, identifying some patients (p53abn-like NSMP) as candidates for treatment analogous to what is given in p53abn tumors. Furthermore, the proposed AI model can be easier to implement in practice (for example, in a cloud-based environment where scanned routine H&E images could be uploaded to a platform for AI assessment), leading to greater impact on patient management. Furthermore, we envision

that an AI algorithm, after appropriate validation, could be utilized on diagnostic biopsy specimens, along with molecular subtype markers (p53, MMR, POLE). This would allow diagnosis of molecular subtype and further classify NSMP into lower risk and higher risk (p53abn-like), with the former patients being candidates for de-escalation of treatment (e.g., simple hysterectomy in the community) and the latter group potentially directed to cancer centers for lymph node assessment, omental sampling and directed biopsies given a higher likelihood of upstaging. It is possible that with further refinement and validation of the algorithm, which can be run in minutes on the diagnostic slide image, that it could take the place of molecular subtype markers, saving time and money.”

R3 Comment 5: The study employed various deep learning-based models for tumor subtype classification, including Vanilla, Histogram-Based, IDaRS, Attention-based, and VLAD. However, how was the selection made for using these specific models? Were any other state-of-the-art models considered for comparison, and what were the reasons for excluding them?

Response: Thank you for comment. We chose these models as representatives of major categories of deep learning algorithms for classification at the time that we conducted the deep learning model construction part of the work on the discovery set. We then locked these models and validated them on the validation cohort and performed extensive genomics and transcriptomics analysis based on these results.

However, due to the fast-paced nature of the AI field, we understand that other state-of-the-art models have been introduced while we were performing validation and genomics studies. Therefore, to address this reviewer’s valid comment, we tested the performance of two state-of-the-art models, namely CLAM and transMIL, and can report that similar findings were observed using these two models.

We have revised the text to include these results and added the detailed results in Supplementary Figure 4 and Supplementary Tables 9 and 10.

R3 Comment 6: In the unsupervised clustering of NSMP patch representations using KimiaNet and UMAP, how sensitive are the results to the choice of dimensionality reduction technique and the number of clusters? Did you conduct any sensitivity analyses to verify the stability of the clustering results?

Response: To assess how the choice of dimensionality reduction technique influences our findings, we experimented with DenseNet121, ResNet50, and Swin. Detailed results of this new experiment are presented in Supplementary Figure 6. As part of the dimensionality reduction technique, we do not specify any pre-defined number of clusters. As a result, the analysis revealed that identified clusters remains consistent, two clusters, across these techniques. Regarding the sensibility analysis, this result is based on unsupervised dimensionality reduction to just demonstrate the difference between NSMP and p53-like NSMP patches using different encodings from DenseNet121, ResNet50, and Swin.

R3 Comment 7: The use of histopathology slides for training and validation raises questions about potential sources of bias and the generalizability of the findings. How do you address the potential heterogeneity in slide staining and imaging quality across different centers, which might impact the model's performance and applicability in other settings?

Response: To address this heterogeneity in slide staining and imaging quality across different centers, we employ color-normalization techniques before applying our model to the data. Having added a new dataset (n = 614 cases) collected from various hospitals across Canada suggests that our models are robust to such variations across sites.

R3 Comment 8: The study validated the AI model's performance on an external validation set from a single center (Vancouver cohort). Considering the variability in tissue processing and data collection across centers, were any efforts made to include additional external validation cohorts from different geographic locations to enhance the model's robustness and generalization?

Response: Done. As suggested, we have confirmed our findings in a dataset collected from various hospitals in Canada. Please see response to comment 7 for further details.

R3 Comment 9: The study identifies a subset of NSMP endometrial cancers with unique clinical outcomes. Can the authors propose plausible biological mechanisms underlying the observed differences in survival outcomes between the p53abn-like NSMP and other NSMP cases? Additional biological investigations may strengthen the study's implications.

Response: We would like to highlight our results pertaining to the genomic characterization of p53abn-like NSMP cases as outlined (section “*Genomic characterization of p53abn-like NSMP cases*”). To understand the biological mechanism underlying the observed differences in survival outcomes between the p53abn-like NSMP and other NSMP cases, we have performed analysis on point mutation data, shallow whole genome sequencing and gene expression profiles. Our results revealed that p53abn-like NSMPs harbor higher copy number variation burden. Furthermore, our gene expression analysis suggested the upregulation of PI3k-Akt, Wnt, and Cadherin signaling pathways both in p53abn-like NSMP and p53abn groups (compared to NSMP). All these results suggest genomic and transcriptomic similarities between the p53abn-like NSMP and p53abn cases and potential defects in the DNA damage repair process as a possible biological mechanism.

As suggested by the reviewer, we have now added more explanation about these findings in the discussion section (also text copied below for convenience):

“Molecular characterization of the identified subtype using sWGS suggests that these cases harbor an unstable genome with a higher fraction of altered genome, similar to the p53abn group but with lesser degree of instability. These results suggest that the identified subgroup based on histopathology images is biologically distinct. Furthermore, our gene expression analysis revealed the upregulation of PI3k-Akt, Wnt, and Cadherin signaling pathways both in p53abn-like NSMP and p53abn groups (compared to NSMP). All these results suggest genomic and transcriptomic similarities between the p53abn-like NSMP and p53abn cases and potential defects in the DNA damage repair process as a possible biological mechanism. However, in spite of the fact that similar gene expression pathways were implicated in both groups and H&E images of both groups as assessed by AI had similarities, expression data analysis revealed minimal overlap between the differentially expressed genes in both p53abn and p53abn-like NSMP ECs compared to other NSMP cases. This observation suggests that they may have different etiologies and warrants further biological interrogation of these groups in future studies.”

REVIEWERS' COMMENTS

Reviewer #1 (Remarks to the Author):

All the comments have been well addressed and the manuscript has been modified appropriately.

I have no further comments.

Reviewer #3 (Remarks to the Author):

1. Noteworthy Results:

The study presents noteworthy results in utilizing AI-based histopathology image analysis to identify a novel subset of endometrial cancers (EC) termed 'p53abn-like NSMP.' This subset exhibits marked differences in progression-free and disease-specific survival compared to the conventional NSMP subtype. The identification of this distinct subgroup through AI is a significant contribution to the field and has potential implications for prognosis and treatment strategies.

2. Significance to the Field:

The work holds substantial significance for the field of endometrial cancer research. The identification of a novel subgroup within the NSMP category, especially through innovative AI-powered histopathology image analysis, adds valuable insights to the molecular subtyping of EC. The findings have potential therapeutic implications and enhance the prognostic value of current molecular subtyping methods.

3. Comparison to Established Literature:

The paper adequately discusses the comparison of the newly identified 'p53abn-like NSMP' subgroup with the established molecular subtypes. Furthermore, a more in-depth comparison with relevant literature on EC molecular subtypes and their clinical outcomes have been provided for the revision.

4. Support for Conclusions:

The conclusions drawn are well-supported by the presented data and analyses. The use of

multiple cohorts for validation adds credibility to the findings.

5. Flaws in Data Analysis, Interpretation, and Conclusions:

The data analysis and interpretation appear robust, and a more detailed discussion on potential biases and limitations in the AI-based histopathology image analysis has been provided in this revision.

6. Methodology:

The methodology, particularly the AI-based histopathology image analysis, has been detailed more comprehensively in this revision.

7. Reproducibility:

The study mentions the use of AI for image analysis, and more details on the availability of the AI model and code explanation have been expanded.

Reviewer #3 (Remarks on code availability):

The code is basically runnable and usable, and the results are reproducible.